# [^18^F]FDG PET/CT Imaging Is Associated with Lower In-Hospital Mortality in Patients with Pyogenic Spondylodiscitis—A Registry-Based Analysis of 29,362 Cases

**DOI:** 10.3390/antibiotics13090860

**Published:** 2024-09-08

**Authors:** Siegmund Lang, Nike Walter, Stefanie Heidemanns, Constantin Lapa, Melanie Schindler, Jonas Krueckel, Nils Ole Schmidt, Dirk Hellwig, Volker Alt, Markus Rupp

**Affiliations:** 1Department of Trauma Surgery, University Hospital Regensburg, 93053 Regensburg, Germany; siegmund.lang@ukr.de (S.L.); melanie.schindler@ukr.de (M.S.);; 2Department of Nuclear Medicine, University Hospital Regensburg, 93053 Regensburg, Germany; 3Nuclear Medicine, Faculty of Medicine, University of Augsburg, 86154 Augsburg, Germany; 4Department of Neurosurgery, University Hospital Regensburg, 93053 Regensburg, Germany

**Keywords:** spondylodiscitis, vertebral osteomyelitis, [^18^F]FDG PET/CT, in-hospital mortality, elderly patients

## Abstract

Background: While MRI is the primary diagnostic tool for the diagnosis of spondylodiscitis, the role of [^18^F]-fluorodeoxyglucose ([^18^F]FDG) PET/CT is gaining prominence. This study aimed to determine the frequency of [^18^F]FDG PET/CT usage and its impact on the in-hospital mortality rate in patients with spondylodiscitis, particularly in the geriatric population. Methods: We conducted a Germany-wide cross-sectional study from 2019 to 2021 using an open-access, Germany-wide database, analyzing cases with ICD-10 codes M46.2-, M46.3-, and M46.4- (‘Osteomyelitis of vertebrae’, ‘Infection of intervertebral disc (pyogenic)’, and ‘Discitis unspecified’). Diagnostic modalities were compared for their association with in-hospital mortality, with a focus on [^18^F]FDG PET/CT. Results: In total, 29,362 hospital admissions from 2019 to 2021 were analyzed. Of these, 60.1% were male and 39.9% were female, and 71.8% of the patients were aged 65 years and above. The overall in-hospital mortality rate was 6.5% for the entire cohort and 8.2% for the geriatric subgroup (*p* < 0.001). Contrast-enhanced (ce) MRI (48.1%) and native CT (39.4%) of the spine were the most frequently conducted diagnostic modalities. [^18^F]FDG PET/CT was performed in 2.7% of cases. CeCT was associated with increased in-hospital mortality (OR = 2.03, 95% CI: 1.90–2.17, *p* < 0.001). Cases with documented [^18^F]FDG PET/CT showed a lower frequency of in-hospital deaths (OR = 0.58, 95% CI: 0.18–0.50; *p* = 0.002). This finding was more pronounced in patients aged 65 and above (OR = 0.42, 95% CI: 0.27–0.65, *p* = 0.001). Conclusions: Despite its infrequent use, [^18^F]FDG PET/CT was associated with a lower in-hospital mortality rate in patients with spondylodiscitis, particularly in the geriatric cohort. This study is limited by only considering data on hospitalized patients and relying on the assumption of error-free coding. Further research is needed to optimize diagnostic approaches for spondylodiscitis.

## 1. Introduction

Spondylodiscitis is a severe musculoskeletal infection that is defined as an infection of the intervertebral disc with potential osteomyelitis of the adjacent vertebral bodies [1]. Despite progress in diagnostics and therapy in recent decades, spondylodiscitis is still very common, and incidence rates have increased in recent years. For Germany, an increase from 10.4 to 14.4/100,000 was reported between 2010 and 2020 [2]. The aging of the population, medical therapy with indwelling devices leading to bloodstream infection and consecutive infection of the spine, and an increasing number of immunocompromised patients are the reasons for the growing incidence rates [3,4,5]. Early diagnosis and treatment of spondylodiscitis are crucial to prevent severe complications, including neurological deficits and sepsis, which can lead to increased mortality. The etiology entails bloodstream infection leading to so-called hematogenous spondylodiscitis. On the other hand, implant-associated vertebral osteomyelitis may occur after surgery with the application of orthopedic implants [6,7]. Patients who undergo surgical treatment for spondylodiscitis face high rates of additional complications, such as persistent infection, implant failure, and the need for reoperation, further complicating the clinical course and outcomes [8,9]. Despite modern diagnostics and therapy, mortality in patients with spondylodiscitis is still quite high and reportedly reaches up to 15–20% [4,10,11]. Patient history is often very unspecific, and diagnosis is not always simple. Standardized diagnostic work-up includes laboratory tests such as white blood cell (WBC) count, serum C-reactive protein (CRP), and blood cultures [12,13]. Recent advancements in diagnostic imaging, particularly in MRI and PET/CT technologies, have significantly improved the detection and management of spondylodiscitis. These advancements have allowed for earlier and more accurate diagnosis, which is essential for reducing the risk of severe outcomes. Hence, standard radiographs of the spine should be performed, as well as magnetic resonance imaging (MRI), which is currently regarded as the preferred primary diagnostic tool. Computed tomography (CT)-guided biopsy as an invasive procedure and microbiological and histological analysis can prove the diagnosis [14,15]. Positron emission tomography/computed tomography with 18F-labelled fluorodeoxyglucose ([^18^F]FDG PET/CT), which is often only available in specialized centers, has been demonstrated to be beneficial in confirming the diagnosis of spondylodiscitis [16]. Despite some disadvantages, such as availability and costs, the sensitivity and specificity to diagnose spondylodiscitis seem similar to those of MRI [16]. Furthermore, [^18^F]FDG PET/CT can easily be offered to patients with pacemakers in whom MRI cannot always be performed [17]. In contrast to MRI, [^18^F]FDG PET/CT provides reliable image quality even in cases with implant-associated vertebral osteomyelitis [18]. In addition to these benefits, [^18^F]FDG PET/CT enables a focused whole-body search of additional infection foci, which is very useful, especially for patients with hematogenous etiology of spondylodiscitis: In patients with *Staphylococcus aureus* bacteremia, [^18^F]FDG PET/CT frequently identified metastatic foci of infection; revealed foci undetected by prior investigations; and led to additional source control procedures, fewer infection relapses, and consequently lower mortality [19]. Intriguingly, the number needed to treat to prevent death in patients with *Staphylococcus aureus* bacteremia has been reported to be 4–8 [^18^F]FDG PET/CT scans [20]. Therefore, this study addresses a critical gap and potential in current diagnostic practices by evaluating the innovative application of [^18^F]FDG PET/CT in detecting spondylodiscitis, particularly in complex clinical scenarios in which traditional methods may fall short.

Specifically, this study aimed to determine (1) the frequency of [^18^F]FDG PET/CT usage in the diagnostic regimen for a specific cohort of hospitalized patients with pyogenic spondylodiscitis. (2) Furthermore, we aimed to compare the diagnostic value of [^18^F]FDG PET/CT in terms of its association with in-hospital mortality with other key diagnostic methods for patients with spondylodiscitis, including MRI, CT-guided biopsy, and endocarditis screening using transesophageal sonography. (3) Lastly, considering the limited availability of [^18^F]FDG PET/CT, this study aimed to identify the patient cohort that would benefit the most from its additional use in terms of the risk of in-hospital mortality, particularly focusing on geriatric patients aged 65 years and above.

## 2. Results

### 2.1. Epidemiological Features and Frequency of Diagnostic Modalities in Patients with Spondylodiscitis

A total of 29,362 hospital admissions were analyzed from 2019 to 2021. The proportion of male patients was 60.1% (n = 17,654), while that of female patients was 39.9% (n = 11,705). N = 21,090 (71.8%) patients were aged 65 years and above (Figure 1). In terms of length of stay, 2066 (7.0%) were short-stay patients, 21,581 (73.5%) were normal-stay patients, and 5715 (19.5%) were long-stay patients. The average length of stay was 22.4 days, with a standard deviation of 19.7. In the patient classification by PCCL (Patient Clinical Complexity Level), 23.6% (n = 6935) were classified as level 0, 10.4% (n = 3041) as level 1, 12.3% (n = 3617) as level 2, 22.6% (n = 6635) as level 3, 20.4% (n = 5991) as level 4, 9.5% (n = 2788) as level 5, and 1.2% (n = 355) as level 6. 

The odds of in-hospital mortality closely mirrored the PCCL categories, with lower categories showing reduced odds, as follows: PCCL 0, OR = 0.11 (95% CI: 0.08–0.14, *p* < 0.001); PCCL 1, OR = 0.26 (95% CI: 0.19–0.34, *p* < 0.001); and PCCL 2, OR = 0.49 (95% CI: 0.40–0.59, *p* < 0.001). As the complexity increased, so did the odds of mortality (PCCL 3—OR = 0.87, 95% CI: 0.78–0.97, *p* = 0.028; PCCL 4—OR = 1.97, 95% CI: 1.80–2.16, *p* < 0.001; PCCL 5—OR = 2.98, 95% CI: 2.68–3.33, *p* < 0.001; PCCL 6—OR = 6.07, 95% CI: 4.81–7.66, *p* < 0.001). These numbers indicate higher mortality risks with increased clinical complexity.

In this Germany-wide cohort, the most conducted diagnostic modalities (listed in Table 1) included contrast-enhanced (ce) MRI of the spine and spinal cord, accounting for 14,137 cases (48.1%), followed by native CT of the spine and spinal cord, with 11,573 cases (39.4%). Transesophageal echocardiography (TEE) was performed in 6948 cases (23.7%). Other notable diagnostic modalities in the work-up of patients with spondylodiscitis included native CT of the skull (3967 cases, 13.5%), native CT of the thorax (1999 cases, 6.8%), and native CT of the abdomen (2025 cases, 6.9%). In sum, 14,873 patients underwent ceCT (50.7%), with ceCT of the thorax being most frequently performed in 5075 cases (17.3%). 

Focusing on [^18^F]FDG PET/CT, a total of 801 patients (2.7%) underwent this diagnostic modality. These included 201 cases (0.7%) of [^18^F]FDG PET/CT torso imaging, 159 cases (0.5%) of [^18^F]FDG PET/CT, skull to tight with low-dose CT for attenuation correction, and 144 cases (0.5%) of [^18^F]FDG PET/CT, skull to thigh with diagnostic CT. Additionally, 172 cases (0.6%) involved [^18^F]FDG PET/CT of the whole body (head to toe) with low-dose CT for attenuation correction and 125 cases (0.4%) with diagnostic, contrast-enhanced CT. Regarding biopsies, a total of 1236 cases (4.2%) underwent open biopsy of the spine, while 1413 cases (4.8%) had percutaneous biopsy of the spine. Another 393 cases (1.3%) underwent percutaneous needle biopsy of the spine; thus, the total number of biopsies performed was 3042 (10.4%).

### 2.2. Analysis of the Association between [^18^F]FDG PET/CT and In-Hospital Mortality among Patients with Spondylodiscitis 

The overall in-hospital mortality rate in the total cohort was 6.5%. In the subgroup of patients who underwent [^18^F]FDG PET/CT, there were 770 survivors and 31 deaths, resulting in a mortality rate of 3.9%. The OR for cases with [^18^F]FDG PET/CT in relation to in-hospital mortality was 0.58 (95% CI: 0.18–0.50; *p* = 0.002). Open biopsy of the spine (OR = 0.86, 95% CI: 0.67–1.09, *p* = 0.196) showed a trend toward an association with reduced odds of in-hospital mortality. Another potential protective factor, with an OR of less than 1.00, was a history of percutaneous needle biopsy of the spine (OR = 0.71, 95% CI: 0.56–0.88, *p* = 0.001).

Procedures associated with increased in-hospital mortality (OR > 1.00) included musculoskeletal ceCT (OR = 2.50; 95% CI: 1.67–3.74), abdominal ceCT (OR = 1.81, 95% CI: 1.65–1.99, *p* < 0.001), pelvic ceCT (OR = 2.16, 95% CI: 1.89–2.46, *p* < 0.001), and thoracic ceCT (OR = 2.00, 95% CI: 1.82–2.20, *p* < 0.001). When accumulated, all ceCT scans showed a significant association with in-hospital mortality (OR = 2.03, 95% CI: 1.90–2.17, *p* < 0.001). A less prominent but significant association was observed for the use of native MRI (OR = 1.14, 95% CI: 1.04–1.25, *p* = 0.001). By contrast, native CT of the spine (OR = 1.01, 95% CI: 0.93–1.10, *p* = 0.715) and ceMRI (OR = 0.96, 95% CI: 0.89–1.05, *p* = 0.137) were not associated with a relevant change in the OR for in-hospital mortality (Figure 2A). 

### 2.3. Diagnostic Modalities and the Role of [^18^F]FDG PET/CT Usage in the Geriatric Patient Cohort, Aged 65 Years and Above

The mortality rate in the geriatric cohort consisting of 21,090 patients was 8.2% (n = 1729), compared with 6.5% in the total cohort (*p* < 0.001). The in-hospital mortality rate in cases of geriatric patients, in whom [^18^F]FDG PET/CT was documented, was 3.9%. The analysis of diagnostic modalities revealed significant differences between the total cohort and patients aged 65 and above (Table 2). In the older cohort, percutaneous bone biopsy of the spine (5.7% vs. 6.2%, *p* < 0.001) and open biopsy of the spine (3.9% vs. 4.2%, *p* < 0.001) were less frequent, while TEE was more frequent (24.2% vs. 22.9%, *p* < 0.001). CT of regions other than the spine was less common in the older group (24.7% vs. 36.3%, *p* < 0.001). However, there were no significant differences in spinal CT frequency (39.3% vs. 39.4%, *p* = 0.402), including ceCT of the spine (5.6% vs. 5.6%, *p* = 0.119). Native MRI was slightly more common in the older cohort (33.5% vs. 32.7%, *p* < 0.001), and ceMRI showed a slight lower frequency (47.5% vs. 48.1%, *p* = 0.001). There was no difference in the frequency of the use of [^18^F]FDG PET/CT (2.7% vs. 2.7%, *p* = 0.771).

In the geriatric cohort, certain diagnostic modalities were associated with increased odds of in-hospital mortality. Pelvic ceCT (OR = 2.06, 95% CI: 1.78–2.37, *p* < 0.001), cumulative ceCT of regions other than the spine (OR = 1.91, 95% CI: 1.77–2.05, *p* < 0.001), thoracic ceCT (OR = 1.88, 95% CI: 1.69–2.08, *p* < 0.001), abdominal ceCT (OR = 1.69, 95% CI: 1.53–1.87, *p* < 0.001), and transesophageal echocardiography (TEE) (OR = 1.42, 95% CI: 1.28–1.56, *p* < 0.001) were all associated with higher in-hospital mortality.

On the other hand, certain diagnostic modalities showed a theoretical protective effect for in-hospital mortality. These included native CT of the spine (OR = 0.69, 95% CI: 0.49–0.94, *p* < 0.001) and percutaneous needle biopsy (OR = 0.72, 95% CI: 0.56–0.91, *p* = 0.005). [^18^F]FDG PET/CT again demonstrated the lowest association with in-hospital mortality of the evaluated modalities (OR = 0.42, 95% CI: 0.27–0.65, *p* = 0.001) (Figure 2B).

## 3. Discussion

Whereas [^18^F]FDG PET/CT was only performed in a minority of cases, its use was associated with a decreased odds ratio for in-hospital mortality among patients with spondylodiscitis compared with those who did not receive this diagnostic procedure. Of note, this effect was even more pronounced in patients aged 65 years and above.

### 3.1. Frequency of Diagnostic Modalities Patients with Spondylodiscitis

In a Germany-wide cohort of 29,362 patients, ceMRI of the spine and spinal cord (48.1%) and native CT of the spine and spinal cord (39.4%) were the most commonly conducted diagnostic modalities, followed by TEE in 23.7% of cases. These findings align with current guidelines, which establish ceMRI as the radiographic gold standard for diagnosing pyogenic spondylodiscitis [21,22], whereas CT offers advantages in quantifying bone loss and plays a crucial role in surgical planning and guided biopsies [23,24]. Percutaneous needle biopsy and open surgical biopsy, conducted in 1.3% and 4.2% of cases, respectively, were relatively infrequent procedures. In the sub-cohort of patients aged 65 or above, biopsies were performed significantly less frequently. Securing tissue samples for pathogen identification is crucial and has been incorporated into several guidelines for spondylodiscitis [21,25,26]. Adding up all the procedures, ceCT scans were conducted in 50.7% of cases. The use of contrast agents plays an important role in improving diagnostic accuracy, aiding in the assessment of lesion extent and characteristics, particularly in cases of spondylodiscitis, in which the detection of infection and abscesses is of utmost importance [27,28].

The availability of both CT and MRI has significantly increased, making these modalities broadly accessible [29,30]. Conversely, [^18^F]FDG PET/CT remains primarily limited to specialized centers [31], a fact that is also evidenced by its employment in a mere 2.7% of cases in the present study. While the low availability may contribute to this usage, a standardized application of [^18^F]FDG PET/CT in infectious diseases, including spondylodiscitis, has not yet been fully established [32,33]. Given its high sensitivity, [^18^F]FDG PET/CT is gaining prominence in the diagnosis and management of spondylodiscitis, particularly in postoperative scenarios involving spinal implants or other implantable devices for which MRI has limitations [34]. A recent review of 36 meta-analyses by Treglia and colleagues demonstrated the good diagnostic performance of [^18^F]FDG PET/CT in detecting various inflammatory and infectious diseases, particularly spondylodiscitis, among other musculoskeletal infections [35]. The authors called for more prospective multicenter studies and cost-effectiveness analyses [35].

### 3.2. Analysis of the Association between [^18^F]FDG PET/CT and In-Hospital Mortality among Patients with Spondylodiscitis 

In this cohort, the use of [^18^F]FDG PET/CT was significantly more frequent in the survival group than in the deceased group, with an odds ratio of 0.58. In comparison, the conduction of native CT of the spine (OR = 1.01) and ceMRI, the preferred imaging standard (OR = 0.96), did not show a relevant association with mortality. While a causal relationship between [^18^F]FDG PET/CT and in-hospital mortality cannot be established, several factors could contribute to this statistical correlation. The importance of early detection of spondylodiscitis for improved clinical outcomes has been emphasized, as delays can lead to fatal outcomes. [36,37]. In their study on 49 patients with spondylodiscitis, Smids et al. found that [^18^F]FDG PET/CT, with 96% sensitivity and 95% specificity, outperformed MRI in early spondylodiscitis detection [38]. After 2 weeks, MRI’s performance (67% sensitivity, 84% specificity) matched that of [^18^F]FDG PET/CT [38]. In line with this finding, Yang et al. reported an intriguing case of spondylodiscitis, highlighting the disease’s cryptic early-stage features, which often lead to delayed diagnosis. They emphasized the utility of [^18^F]FDG PET/CT and MRI, particularly for patients with persistent symptoms, in improving diagnostic accuracy [39]. [^18^F]FDG PET/CT also plays a valuable role in diagnosing postoperative, implant-associated vertebral osteomyelitis [7]. Along these lines, Dauchy et al. found that [^18^F]FDG PET/CT had a sensitivity of 85.7%, a specificity of 82.6%, a positive predictive value of 60.0%, and a negative predictive value of 95.0% for diagnosing spinal infection relapse [40]. Berrevoets et al. conducted a retrospective study of 184 *Staphylococcus aureus* bacteremia (SAB) patients, demonstrating that the use of [^18^F]FDG PET/CT significantly reduced three-month mortality from 32.7% to 12.1% [19]. In their study, [^18^F]FDG PET/CT led to treatment modifications in 74.7% of high-risk patients and detected unrelated diseases in 7.1% of cases [19]. Similarly, Yildiz et al. found that the use of [^18^F]FDG PET/CT led to a significant reduction in mortality rates among 102 high-risk SAB patients [41]. These findings suggest that [^18^F]FDG PET/CT aids in the early detection of pyogenic spondylodiscitis, metastatic septic infection, and postoperative implant-associated vertebral osteomyelitis, optimizing diagnostics and potentially reducing the risk of fatal disease progression. 

Another explanation for our findings could be that [^18^F]FDG PET/CT is primarily performed in specialized centers due to its limited availability. These centers likely have advanced protocols for managing infectious diseases and offer close interdepartmental cooperation, often based on multidisciplinary boards [42]. As a result, patients who undergo [^18^F]FDG PET/CT may be more likely to benefit from the specialized care and expertise available at these centers, rather than the [^18^F]FDG PET/CT itself being a protective factor.

### 3.3. Diagnostic Modalities and the Role of [^18^F]FDG PET/CT Usage in the Geriatric Patient Cohort, Aged 65 Years and Above

The significant proportion (71.8%) of patients aged 65 years or above in the current study is mirrored in up-to-date epidemiological data [2,43]. It is widely recognized that advanced age significantly contributes to the risk of mortality in patients with spondylodiscitis [44]. The current study revealed a pronounced theoretical protective effect of [^18^F]FDG PET/CT on in-hospital mortality in patients with spondylodiscitis, particularly in the geriatric cohort aged 65 years and above (OR = 0.42). We did not find significant differences in the frequencies of [^18^F]FDG PET/CT and ceCT between the total cohort and the elderly. One explanation for this age-specific effect may be the advantage of [^18^F]FDG PET/CT in not relying on (potentially) nephrotoxic contrast agents; a common complication of ceCT is contrast-induced nephropathy (CIN) and, consequently, acute kidney injury (AKI) [40]. In hospitalized patients, AKI is estimated to occur in 2–7% of cases with risk factors like age, sepsis, surgeries, comorbidities, and specific medical interventions, culminating in a mortality rate of up to 80% among ICU patients [45,46]. Elderly patients, often suffering from multiple comorbidities, including diabetes, and pre-existing renal insufficiency, are especially prone to CIN and AKI [47]. In our study, the use of contrast agents in CT diagnostic modalities was associated with an increased odds ratio for in-hospital mortality in elderly patients (OR = 1.91), whereas native CT of the spine showed a reduced OR of 0.69 (*p* < 0.05).

The diagnostic advantages of [^18^F]FDG PET/CT may be particularly beneficial for the geriatric population, as this imaging modality can help prevent delays in diagnosis and, consequently, unfavorable outcomes. As demonstrated by Dubost et al., the time to diagnosis tends to be longer in elderly patients [48]. This delay could not only contribute to fatal outcomes in spondylodiscitis cases but also increase the risk of metastatic infection, as observed in patients with bacteremia induced by methicillin-sensitive *Staphylococcus aureus* [49]. 

The All Patient Refined Diagnosis Related Groups (APR-DRG) classification system is widely utilized for predicting patient outcomes, including inpatient mortality, by categorizing patients on the basis of the severity of their illness and risk of mortality [50]. De Marco et al. found that the APR-DRG classification system has varying effectiveness in predicting inpatient mortality, with strong performance for myocardial infarction (c-statistic 0.91) but lower accuracy for conditions like stroke (c-statistic 0.68) [51]. The inclusion of age and sex improved the model’s performance, although the APR-DRG system alone does not fully capture all relevant risk factors and thus might be limited, particularly in complex cases such as spondylodiscitis. Future studies should consider integrating additional clinical parameters and advanced diagnostic tools, such as [^18^F]FDG PET/CT, to enhance the prognostic capabilities of the APR-DRG system for inpatient mortality in specific patient populations.

The radiological identification of spondylodiscitis in the elderly is especially challenging due to the high prevalence of infection-mimicking degenerative and inflammatory findings, such as Modic changes [52]. Furthermore, the insidious course of spondylodiscitis caused by low-virulent pathogens, which are common in the geriatric population, may contribute to a diagnostic delay [4]. Importantly, elderly multimorbid patients are particularly susceptible to metastatic infection, where [^18^F]FDG PET/CT can fully exploit its original advantage of whole-body-focused identification [34,53]. The current study does not consider individual risk factors, such as comorbidities, concurrent diseases, and complications within the evaluated patient cohort. These factors and their association with in-hospital mortality remain to be investigated on a Germany-wide scale to fully understand their impact.

### 3.4. Limitations

This study has several limitations. In general, registry studies are commonly not accepted by design to allow causative relationships to be made. First, the study only considered data on hospitalized patients; however, spondylodiscitis is usually managed as an inpatient procedure, and the use of [^18^F]FDG PET/CT in an ambulatory setting is rare. Furthermore, the analysis is based on registry data consisting of ICD-10 diagnosis codes from all German medical institutions. Although the correct coding of diagnosis can be assumed since DRG lump-sum payment relies on it (and this is strictly controlled by the Medical Service of Health Funds), it must be acknowledged that the accuracy and completeness of the data rely on the documentation quality of healthcare providers. Hence, patients with atypical or undiagnosed spondylodiscitis who were captured in the InEK database may have been missed. The InEK GmbH does not guarantee that the InEK Daten Browser is entirely error-free. Additionally, although this study aimed to compare the diagnostic value of [^18^F]FDG PET/CT with other key diagnostic methods, such as MRI, CT-guided biopsy, and transesophageal sonography, the analysis may not have accounted for all relevant diagnostic modalities and factors that may have an impact on in-hospital mortality. Notably, the frequency of plain radiographs was not evaluated in the current study. Potential confounding factors, such as patient comorbidities and the availability of specialized care, may influence the outcomes observed in this study. Selection bias is also a concern, as patients undergoing [^18^F]FDG PET/CT may have received more comprehensive care in specialized centers.

Furthermore, it remains unclear where the difference lies between the 201 cases with PET/CT of the entire trunk and the 159 cases with low-dose CT or the 144 cases with contrast-enhanced CT. A possible explanation is the presence of PET scans without corresponding CT, although this is uncommon. Lastly, this cross-sectional study was conducted using Germany-wide data from Germany, and the results may not be applicable to other healthcare systems or regions where access to this diagnostic modality is different.

## 4. Materials and Methods

We conducted a cross-sectional Germany-wide study with pooled data from 2019 through 2021. In accordance with Section 17b of the German Hospital Financing Act, a universal, performance-based, and flat-rate remuneration system has been introduced for general hospital services. The basis for this is the German Diagnosis Related Groups system (G-DRG system), whereby each inpatient case of treatment is remunerated using a corresponding DRG lump-sum payment. The “Institut für das Entgeltsystem im Krankenhaus GmbH” (InEK, Siegburg, Germany) provides detailed data on the main diagnoses (based on ICD-10 codes), age and gender distribution, length of hospital stays, reasons for discharge (including “death”), coded concomitant diagnoses (based on ICD-10 codes), and coded procedures (based on OPS-codes). The OPS acronym stands for “Operationen- und Prozedurenschlüssel” (Operation and Procedure Classification System), which is a German coding system used to document surgical procedures and other interventions performed during a hospital stay. The data InEK provides are collected from all general hospitals in Germany that are subject to the DRG (Diagnosis-Related Group) system. This includes approximately 1900 hospitals across the country.

Data linkage was performed using the ICD-10 codes as identifiers across the datasets provided by InEK. This enabled the integration of information on diagnoses, procedures, and patient outcomes into a comprehensive dataset for analysis.

The InEK browser provides data back to the year 2019. For the following comprehensive analysis, all cases from 2019 through 2021 with ICD-10 codes M46.2-, M46.3-, and M46.4- (spondylodiscitis diagnoses, specifically ‘Osteomyelitis of vertebrae’, ‘Infection of intervertebral disc (pyogenic)’, and ‘Discitis unspecified’) were extracted and pooled. The InEK GmbH does not guarantee that the InEK Daten Browser is entirely free of inconsistencies. However, these are continuously managed through systematic data validation protocols.

The analysis was conducted on comorbidities that were documented in at least 1.0% of cases. The primary outcome parameter was the event of in-hospital mortality. To identify potential risk factors for in-hospital mortality, the prevalence of comorbidities, concomitant injuries (each by ICD-10 code), and procedures (by OPS codes) were compared between two groups: Group A (Survival Group) and Group B (Deceased Group). In the next step, cases with documented surgical procedures were further divided into these two groups to investigate potential risk and protective factors for in-hospital mortality in surgically treated patients. To answer research question number 3, only patients aged 65 years and above were evaluated and again divided into a survival group and a deceased group.

### 4.1. Ethics Statement

Informed consent and investigational review board (IRB) approval were not required for this cross-sectional study, as it used aggregated data from an anonymous, de-identified, administrative database.

### 4.2. Statistical Analysis

The data were analyzed using SPSS statistics version 29.0 (IBM, SPSS Inc., Armonk, NY, USA). The data are presented as total numbers and as percentages of the total. The Chi-square test was used to statistically compare groups (A and B). To analyze the association between potential risk factors and the outcome parameter of in-hospital mortality, odds ratios (ORs) with 95% confidence intervals (95%CI) were calculated for the previously identified factors.

## 5. Conclusions

Although rarely performed, [^18^F]FDG PET/CT was significantly associated with reduced in-hospital mortality rates in patients with spondylodiscitis, particularly in patients aged 65 and above. While MRI and CT scans remain the primary diagnostic tools, the use of contrast agents in CT scans was associated with increased in-hospital mortality, necessitating careful consideration, especially in patients with pre-existing conditions. Future prospective studies should aim to further investigate the potential benefits of incorporating [^18^F]FDG PET/CT into the diagnostic approach for patients with spondylodiscitis.

Future research should focus on optimizing the use of [^18^F]FDG PET/CT in spondylodiscitis by determining the best timing and patient selection criteria. Larger multicenter studies are needed to confirm its benefits and assess its cost-effectiveness, which could lead to broader adoption in clinical practice for improving outcomes for patients with spondylodiscitis, particularly those who are elderly or have multiple comorbidities. Investigating the cost-effectiveness and accessibility of [^18^F]FDG PET/CT in different healthcare settings will also be crucial in optimizing its clinical utility.

## Figures and Tables

**Figure 1 antibiotics-13-00860-f001:**
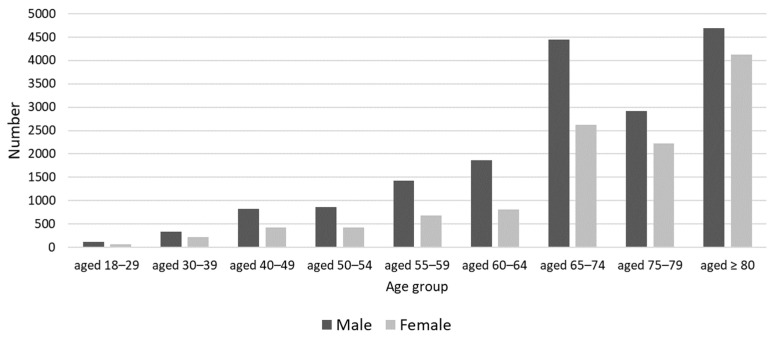
Demographic overview: Age and gender distribution of 29,362 patients with spondylodiscitis included in the current study.

**Figure 2 antibiotics-13-00860-f002:**
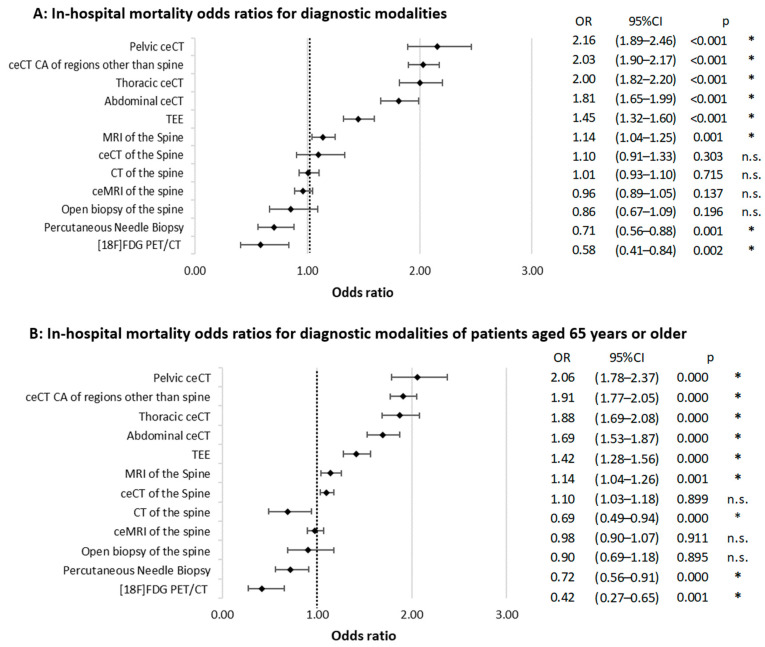
Odds ratios for in-hospital mortality across diagnostic modalities. Part (**A**) presents data for the entire patient cohort, while Part (**B**) focuses on patients aged 65 and older. Whiskers denote the 95% confidence interval (CI), and the dotted line marks an OR of 1.00. *p*-values are derived from Chi-squared tests. “*” indicates statistical significance (*p* < 0.05). “n.s.” denotes “not significant”.

**Table 1 antibiotics-13-00860-t001:** Diagnostic modalities.

OPS Code	Procedure	Absolute Number	Percentage of All Cases	Cumulative Procedure	Accumulated Number	Percentage of All Cases of Accumulated Procedures
1-480.4	Percutaneous Bone Biopsy: Spine	393	1.3%	Percutaneous Bone Biopsy: Spine	1806	6.2%
1-481.4	Non-incisional Bone Biopsy: Spine	1413.0	4.8%			
1-503.4	Incisional Bone Biopsy: Spine	1236	4.2%	Open biopsy of the Spine	1236	4.2%
3-052	Transesophageal Echocardiography (TEE)	6948	23.7%	TEE	6734	22.9%
3-200	Native CT: Skull	3967	13.5%	CT other regions than Spine	10,664	36.3%
3-201	Native CT: Neck	180	0.6%			
3-202	Native CT: Thorax	1999	6.8%			
3-205	Native CT: Musculoskeletal	828	2.8%			
3-206	Native CT: Pelvis	1339	4.6%			
3-207	Native CT: Abdomen	2025	6.9%			
3-20x	Other Native CT	326	1.1%			
3-203	Native CT: Spine and Spinal Cord	11,573	39.4%	CT of the Spine	11,573	39.4%
3-220	contrast-enhanced (ce) CT: Skull	696	2.4%	ceCT other regions than Spine	14,873	50.7%
3-221	ceCT: Neck	796	2.7%			
3-222	ceCT: Thorax	5075	17.3%			
3-225	ceCT: Abdomen	5813	19.8%			
3-226	ceCT: Pelvis	2303	7.8%			
3-227	ceCT: Muskuloskelettal System	190	0.6%			
3-223	ceCT: Spine and Spinal Cord	1669	5.7%	ceCT of the Spine	1669	5.6%
3-742		Torso imaging	201	0.7%	[18F]FDG PET/CT	801
3-752.0	Skull to thigh: Low-dose CT	159	0.5%	[18F]FDG PET/CT Native MRI	801 9597	2.7%
3-752.1	Skull to thigh: Diagnostic CT	144	0.5%			
3-753.0	Whole-body: Low-dose CT	172	0.6%			
3-753.1	Whole-body: Diagnostic CT	125	0.4%			
3-802	Native MRI: Spine and Spinal Cord	9597	32.7%			
3-823	ceMRI: Spine and Spinal Cord	14,137	48.1%	ceMRI	14,137	32.7%

**Table 2 antibiotics-13-00860-t002:** Differences in frequencies of diagnostic modalities.

	Total Cohort		Aged 65 Years and Above		
	29,362		21,090		
Summed Procedure	Number	Percentage	Number	Percentage	Chi Squared Test; *p*-Value
Percutaneous Bone Biopsy: Spine	1806	6.2%	1211	5.7%	<0.001
Open biopsy of the spine	1236	4.2%	817	3.9%	<0.001
TEE	6734	22.9%	5105	24.2%	<0.001
CT of other regions than the spine	10,664	36.3%	5206	24.7%	<0.001
CT of the spine	11,573	39.4%	8281	39.3%	0.402
ceCT other than the spine	14,683	50.0%	10,594	50.2%	0.217
ceCT of the spine	1669	5.6%	1171	5.6%	0.119
[^18^F]FDG-PET/CT	801	2.7%	579	2.7%	0.771
Native MRI	9597	32.7%	7072	33.5%	<0.001
ceMRI	14,137	48.1%	10,021	47.5%	0.001

## Data Availability

The original data presented in the study are openly available in the open registry “InEK Datenbrowser” https://www.g-drg.de/datenlieferung-gem.-21-khentgg/inek-datenbrowser, accessed on 15 April 2023.

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
