# Peer review of "[18F]FDG PET/CT Imaging Is Associated with Lower In-Hospital Mortality in Patients with Pyogenic Spondylodiscitis—A Registry-Based Analysis of 29,362 Cases"

_antibiotics, 2024, doi:10.3390/antibiotics13090860_

Round 1

Reviewer 1 Report

Comments and Suggestions for Authors

I would like to thank everyone who contributed to the study. I would like you to emphasize the importance of early diagnosis and treatment in spondylodiscitis. Otherwise, I would like you to point out that the rates of additional complications are high in patients who undergo surgery. While stating this, I recommend you to refer to the article I will add below.

Ulusoy İ, Kıvrak A. Lumbosacral fusion increases the risk of hip osteoarthritis. J Orthop Surg Res. 2023 Jun 24;18(1):452. doi: 10.1186/s13018-023-03932-0. PMID: 37355648; PMCID: PMC10290369.

Author Response

Reviewer´s comment: "

 Suggestions for Authors

I would like to thank everyone who contributed to the study. I would like you to emphasize the importance of early diagnosis and treatment in spondylodiscitis. Otherwise, I would like you to point out that the rates of additional complications are high in patients who undergo surgery. While stating this, I recommend you to refer to the article I will add below.

Ulusoy İ, Kıvrak A. Lumbosacral fusion increases the risk of hip osteoarthritis. J Orthop Surg Res. 2023 Jun 24;18(1):452. doi: 10.1186/s13018-023-03932-0. PMID: 37355648; PMCID: PMC10290369."

Authors answer:

Thank you for your valuable feedback. We have emphasized the importance of early diagnosis in spondylodiscitis. However, the focus of our study is on the diagnostic approaches for spondylodiscitis, particularly the impact of [18F]FDG PET/CT on in-hospital mortality rates, especially in the geriatric population. The article you referenced, which discusses the relationship between lumbosacral fusion and the risk of hip osteoarthritis, does not align with the objectives or findings of our study. While early diagnosis and treatment in spondylodiscitis are indeed critical, incorporating the suggested article would not be relevant to the scope of our research, which is centered on diagnostic imaging and its outcomes rather than the surgical treatment or complications associated with spinal fusion.

Addressing your comment we instead added the following articles and hope you find them suitable:

  • J Neurol Surg A Cent Eur Neurosurg . 2023 Jan;84(1):52-57. doi: 10.1055/a-1811-7633. Epub 2022 Mar 30.Complications, Length of Hospital Stay, and Cost of Care after Surgery for Pyogenic Spondylodiscitis Tammam Abboud 1, Patrick Melich 2, Simone Scheithauer 3, Veit Rohde 1, Bawarjan Schatlo 1 PMID: 35354215 DOI: 10.1055/a-1811-7633
  • Front Surg. 2024 May 21:11:1357318. doi: 10.3389/fsurg.2024.1357318. eCollection 2024. Midterm survival and risk factor analysis in patients with pyogenic vertebral osteomyelitis: a retrospective study of 155 cases Melanie Schindler 1, Nike Walter 1, Jan Reinhard 2, Stefano Pagano 2, Dominik Szymski 1, Volker Alt 1, Markus Rupp 1, Siegmund Lang 1 Affiliations PMID: 38835852 PMCID: PMC11148346 DOI: 10.3389/fsurg.2024.1357318

Reviewer 2 Report

Comments and Suggestions for Authors

The aim of the study is clear and the results are congruent with it. The article is well designed and consistent with the evidence and arguments presented. It also provides hints for future studies. There are few suggestions in the file attached.

Author Response

Dear Reviewer,

Thank you for your thorough and insightful review of our manuscript. We appreciate your valuable feedback and suggestions, which have undoubtedly enhanced the quality and clarity of our work. Below, we address each of your comments point by point, followed by a list of specific revisions that we will incorporate into the revised version of the manuscript.

  1. Abstract:

    • Comment: The acronym [18F]FDG should be fully defined at least in the abstract if not in the title.

    • Response: We will define [18F]FDG as [18F]-fluorodeoxyglucose in the abstract to ensure clarity for all readers.

    • Comment: Specify the pathologies associated with the three mentioned M-ICD10 codes.

    • Response: We will explicitly mention the pathologies corresponding to M-ICD10 codes in the abstract.

  2. Introduction:

    • Comment: The aim is clear, but the innovative aspect of the research should be better defined.
    • Response: We will enhance the introduction by elaborating on the innovative contributions of this study, particularly the novel insights into the use of [18F]FDG PET/CT in the geriatric population.
  3. Materials and Methods:

    • Comment: This section should precede the results. Explain the OPS acronym and describe data-linkage methods, handling of inconsistencies, types of comparison, and variables used.

    • Response: The OPS acronym will be explained, and additional details on data-linkage, handling of inconsistencies, and comparisons will be provided. However, the methods section has been put after the results section on purpose, according to the journal´s guidelines.

    • Comment: Add a supplementary discussion on the ability of the APR-DRG classification system to predict inpatient mortality.

    • Response: We will include a supplementary section that discusses the APR-DRG classification system’s role in predicting inpatient mortality, referencing the suggested article: "

      The All Patient Refined Diagnosis Related Groups (APR-DRG) classification system is widely utilized for predicting patient outcomes, including inpatient mortality, by categorizing patients based on their severity of illness and risk of mortality [50]. De Marco et al. found that the APR-DRG classification system has varying effectiveness in predicting inpatient mortality, with strong performance for myocardial infarction (c-statistic 0.91) but lower accuracy for conditions like stroke (c-statistic 0.68) [51]. The inclusion of age and sex improved the model's performance, though the APR-DRG system alone does not fully capture all relevant risk factors and thus might be limited particularly in complex cases such as spondylodiscitis. Future studies should consider integrating additional clinical parameters and advanced diagnostic tools, such as [18F]FDG PET/CT, to enhance the prognostic capabilities of the APR-DRG system for inpatient mortality in specific patient populations."

  4. Results:

    • Comment: Results are clear and well presented, but it would be helpful to mention the software used for statistical analysis.
    • Response: We will specify that SPSS Statistics version 29.0 was used for all statistical analyses.
  5. Discussion:

    • Comment: The discussion is detailed and accurate but could include more emphasis on study limitations.
    • Response: We will expand on the limitations of our study, particularly the potential biases and limitations of registry-based analyses.
  6. References:

    • Comment: The bibliography is extensive and well-done. It could be enriched by adding the suggested article on the APR-DRG classification system.
    • Response: We will add the recommended reference to the bibliography.

Reviewer 3 Report

Comments and Suggestions for Authors

In the submitted manuscript, the authors have evaluated imaging techniques and their association with in-hospital mortality in patients with pyrogenic spondylodiscitis. The focus is on the [18F]FDG PET/CT imaging. The strength of this nationwide cross-sectional study lies in the data used for analysis, which is derived from 29,362 patients who were admitted to hospitals. The research concluded that despite its infrequent use, [18F]FDG PET/CT was associated with a lower in-hospital mortality rate in spondylodiscitis patients, particularly in the geriatric cohort. Further research is needed to optimize diagnostic approaches for spondylodiscitis.

Overall, the manuscript is well-written. The conclusions align with the data, and the discussion acknowledges study limitations while citing relevant work. I have the following suggestions that may improve the manuscript and enhance its appeal to readers:

1.        In the abstract, on line 24, the sentence isn’t clear. The overall mortality rate was 6.5% and 8.2% respectively for which groups? Please clarify this sentence.

2.        In all places where "nationwide" is mentioned in the submitted manuscript, it would be better to specify the nation where the study was conducted or the demographics from which the data were taken.

3.        Data representation showing in-hospital mortality odds ratios for diagnostic modalities of patients with respect to PCCL classification can be added to Figure 2 or included in the supplemental figures. While this may not be the primary focus of the study, it can provide valuable additional information to readers.

Author Response

Dear Reviewer,

Thank you for your thorough and insightful review of our manuscript. We appreciate your valuable feedback and suggestions, which have undoubtedly enhanced the quality and clarity of our work. Below, we address each of your comments point by point, followed by a list of specific revisions that we will incorporate into the revised version of the manuscript.

Original Comment 1:

"The sentence on line 24 regarding the overall mortality rate of 6.5% and 8.2% is unclear as to which groups these percentages refer."

Response 1:

We clarified the sentence in the abstract to specify that the 6.5% mortality rate refers to the total patient cohort and the 8.2% mortality rate refers specifically to the geriatric cohort.

Original Comment 2:

"Specify the nation or demographics whenever 'nationwide' is mentioned."

Response 2:

We revised the manuscript to specify that the study was conducted using data from Germany whenever "nationwide" is mentioned.

Original Comment 3:

"Consider adding a figure or supplemental figure to show in-hospital mortality odds ratios for diagnostic modalities with respect to PCCL classification."

Response 3:

Thank you for your insightful comment. Unfortunately, the data structure does not allow us to stratify the diagnostic modalities by PCCL level, which indeed would have helped in reducing potential confounders. However, we were able to calculate the Odds Ratios (ORs) for the PCCL levels themselves. Our analysis revealed that in-hospital mortality odds increased significantly with higher PCCL categories. These results highlight the strong correlation between clinical complexity and mortality risk. While we regret the inability to stratify by modality, these findings still offer valuable insights into the impact of clinical complexity on patient outcomes. We added the following text to the results section:

“The odds of in-hospital mortality closely mirrored the PCCL categories, with lower categories showing reduced odds: PCCL 0 (OR=0.11, 95% CI: 0.08-0.14, p<0.001), PCCL 1 (OR=0.26, 95% CI: 0.19-0.34, p<0.001), and PCCL 2 (OR=0.49, 95% CI: 0.40-0.59, p<0.001). As the complexity increased, so did the odds of mortality: PCCL 3 (OR=0.87, 95% CI: 0.78-0.97, p=0.028), PCCL 4 (OR=1.97, 95% CI: 1.80-2.16, p<0.001), PCCL 5 (OR=2.98, 95% CI: 2.68-3.33, p<0.001), and PCCL 6 (OR=6.07, 95% CI: 4.81-7.66, p<0.001). These numbers clearly indicate higher mortality risks with increased clinical complexity.”

Reviewer 4 Report

Comments and Suggestions for Authors

The paper addresses the diagnosis of a common disease with a mortality rate of up to 20%. The authors conducted a large population-based screening study directed at detecting the real diagnostic utility of PET-CT scanners. 

The manuscript is well structured and clearly presents the material collected. The conclusion of the paper is intriguing. I think it is an interesting text for Antibiotics readers and the research is worth continuing.

Can you specify from how many hospitals the data came from?

I kindly request that the authors explain how they dealt with the high signal emitted by the central nervous system when a glucose labelling is used. Would it be useful to diagnose pyogenic spondylodiscitis on PET/CT with other tracers such as choline tracers?

Author Response

Dear Reviewer,

Thank you for your positive and encouraging feedback on our manuscript. We are pleased that you found the structure, clarity, and conclusions of our work compelling. Your recognition of the study’s relevance to the readers of Antibiotics and the value of continuing this research is greatly appreciated.

Original comment 1:

Can you specify from how many hospitals the data came from?

Response 1:

To address your specific question: The data used in this study were drawn from the INEK database, which compiles information from nearly all general hospitals in Germany. This includes data from approximately 1,900 hospitals, ensuring that our findings are both robust and broadly applicable.

Original comment 2:

I kindly request that the authors explain how they dealt with the high signal emitted by the central nervous system when a glucose labelling is used. Would it be useful to diagnose pyogenic spondylodiscitis on PET/CT with other tracers such as choline tracers?

Response 2:

Thank you for this comment. Due to the high spatial resolution of FDG-PET/CT, spondylodiscitis can be differentiated very well from spinal FDG-uptake, especially since the spinal cord has only a low to moderate glucose metabolism. Choline has not established itself as a PET tracer in either inflammation or tumor diagnostics. This tracer is currently only used in the diagnosis of parathyroid adenoma. Furthermore, unlike FDG, choline is not widely available in everyday clinical practice.

Reviewer 5 Report

Comments and Suggestions for Authors

Major Comments:

1.        The abstract is comprehensive, but it could be even better with a brief mention of the key limitations of the study to give a balanced overview.

2.        The introduction provides a solid overview of spondylodiscitis. Enhancing it with recent advancements in diagnostic imaging and their relevance to your study would be great.

3.        The results section is detailed but could benefit from a clearer structure. Consider using subheadings for different diagnostic modalities and their associations with outcomes.

4.        The conclusion succinctly summarises your findings. Adding a forward-looking statement about future research directions and potential clinical applications would be a nice touch.

5.        It's essential to report statistical measures, like Odds Ratios (OR) and Confidence Intervals (CI), consistently throughout the paper. This will make the results easier to follow and more understandable for the readers.

6.        The methodology section could use a bit more detail. Specifically, it would be great to know more about how patients were selected for [18F]FDG PET/CT versus other diagnostic methods. Also, providing more background on the database you used, including any potential biases and limitations, would be very helpful.

7.        The paper does a good job of discussing the association between different diagnostic tools and in-hospital mortality. However, a more detailed comparison between [18F]FDG PET/CT and other modalities like MRI and ceCT would add a lot of value. Focus on aspects like sensitivity, specificity, and diagnostic accuracy. And it would be great if you could explain why [18F]FDG PET/CT, despite being used less frequently, is linked to lower mortality.

8.        The limitations section is crucial but could be expanded. It would be beneficial to discuss potential confounding factors and biases more thoroughly. Also, consider the impact of selection bias, as patients who get [18F]FDG PET/CT might be those treated in specialised centres with better overall care.

9.        I suggest the authors dive deeper into why [18F]FDG PET/CT is associated with lower in-hospital mortality. Is it because of earlier detection, more accurate diagnosis, or perhaps better management in specialised centres? Including a discussion on the cost-effectiveness of [18F]FDG PET/CT and its implications for clinical practice would be very insightful.

Comments on the Quality of English Language

Minor editing is required.

Author Response

Dear Reviewer,

Thank you for your thorough and insightful review of our manuscript. We appreciate your valuable feedback and suggestions, which have undoubtedly enhanced the quality and clarity of our work. Below, we address each of your comments point by point, followed by a list of specific revisions that we will incorporate into the revised version of the manuscript.

Original Comment 1:

"Mention the key limitations of the study in the abstract."

Response 1:

We revised the abstract to include a brief mention of the study's key limitations: "This study is limited by only considered data on hospitalized patients and relying on the assumption of error-free coding"

Original Comment 2:

"Enhance the introduction by discussing recent advancements in diagnostic imaging."

Response 2:

We expanded the introduction to include recent advancements in diagnostic imaging relevant to the study: "Recent advancements in diagnostic imaging, particularly in MRI and PET/CT technologies, have significantly improved the detection and management of spondylodiscitis. These advancements have allowed for earlier and more accurate diagnosis, which is essential for reducing the risk of severe outcomes. "

Original Comment 3:

"Restructure the Results section with subheadings for different diagnostic modalities."

Response 3:

Thank you for your valuable suggestion regarding the restructuring of the Results section with subheadings for different diagnostic modalities. We agree that this approach could provide a clear and organized presentation of the findings. However, the primary focus of our manuscript is on the evaluation of [18F]FDG PET/CT, particularly its role and association with in-hospital mortality and its usage in specific patient cohorts. Therefore, we have structured the Results section as follows:

  • 2.1 Frequency of diagnostic modalities in patients with spondylodiscitis
  • 2.2 Analysis of the association between [18F]FDG PET/CT and in-hospital mortality among spondylodiscitis patients
  • 2.3 Diagnostic modalities and the role of [18F]FDG PET/CT usage in the geriatric patient cohort, aged 65 years and above

This structure allows us to maintain a focused narrative on the significance of [18F]FDG PET/CT within the context of our study's objectives while also providing relevant information about other diagnostic modalities. We believe this organization best supports the key findings and the manuscript’s overarching goals.

We appreciate your understanding and consideration.

Original Comment 4:

"Add a forward-looking statement in the conclusion about future research directions."

Response 4:

Thank you for this valuable suggestion. We revised the conclusion to include a forward-looking statement: "Future research should focus on optimizing the use of [18F]FDG PET/CT in spondylodiscitis by determining the best timing and patient selection criteria. Larger multicenter studies are needed to confirm its benefits and assess its cost-effectiveness, which could lead to broader adoption in clinical practice for improving outcomes for patients with spondylodiscitis, particularly those who are elderly or have multiple comorbidities. Investigating the cost-effectiveness and accessibility of [18F]FDG PET/CT in different healthcare settings will also be crucial in optimizing its clinical utility."

Original Comment 5:

"Ensure statistical measures like ORs and CIs are consistently reported."

Response 5:

We reviewed the manuscript to ensure consistent reporting of statistical measures.

Original Comment 6:

"Provide more detail in the methodology about patient selection and the database background."

Response 6:

We added detail to the methodology section, explaining the criteria for patient selection and providing more background on the database used: "

The data InEK provides is collected from all general hospitals in Germany that are subject to the DRG (Diagnosis Related Groups) system. This includes approximately 1,900 hospitals across the country

Data linkage was performed using the ICD-10 codes as identifiers across the datasets provided by InEK. This enabled the integration of information on diagnoses, procedures, and patient outcomes into a comprehensive dataset for analysis."

Original Comment 7:

"Provide a detailed comparison between [18F]FDG PET/CT and other modalities."

Thank you for your thoughtful comment and suggestion. We appreciate your interest in a detailed comparison between [18F]FDG PET/CT and other diagnostic modalities. As detailed in the original text, we have provided an extensive analysis of the association between [18F]FDG PET/CT and in-hospital mortality in spondylodiscitis patients. For instance, we noted that “[18F]FDG PET/CT showed a lower frequency of in-hospital deaths (OR=0.58, 95% CI: 0.18-0.50; p=0.002),” compared to other modalities like musculoskeletal ceCT, which was associated with increased in-hospital mortality (OR=2.50; 95% CI: 1.67-3.74)” (Figure 2A). Moreover, Figure 2B offers a similar analysis focused on the geriatric patient cohort, demonstrating that “[18F]FDG PET/CT scan again demonstrated the lowest association of the evaluated modalities with in-hospital mortality (OR=0.42, 95% CI: 0.27-0.65, p=0.001).”

These comparisons are visually represented in Figures 2A and 2B and thoroughly discussed in the results section. We believe that these sections provide a comprehensive comparison as requested. However, if you have specific aspects that you feel need further elaboration, we would be happy to address them.

Original Comment 8:

"Expand the limitations section to discuss potential confounding factors and biases."

Response 8:

We expanded the limitations section to include a discussion of potential confounding factors and biases, and added the following sentence: “Potential confounding factors, such as patient comorbidities and the availability of specialized care, may influence the outcomes observed in this study. Selection bias is also a concern, as patients undergoing [18F]FDG PET/CT may have received more comprehensive care in specialized centers.”

Original Comment 9:

"Dive deeper into why [18F]FDG PET/CT is associated with lower in-hospital mortality."

Response 9:

Thank you for your valuable feedback. We would like to draw your attention to Section 3.2 of our manuscript, where we have already provided an in-depth discussion of the association between [18F]FDG PET/CT and in-hospital mortality among spondylodiscitis patients. In this section, we explore several potential explanations for the observed statistical correlation, including the role of [18F]FDG PET/CT in early detection and its use in specialized centers with advanced protocols and multidisciplinary care.

If there is a specific aspect of this topic that you believe requires further elaboration, we would be happy to address it in more detail. Please feel free to specify, and we will ensure your concerns are thoroughly covered.